# Characterization of Two New Multidrug-Resistant Strains of *Mycobacterium smegmatis*: Tools for Routine In Vitro Screening of Novel Anti-Mycobacterial Agents

**DOI:** 10.3390/antibiotics8010004

**Published:** 2019-01-02

**Authors:** Patrick K. Arthur, Vincent Amarh, Precious Cramer, Gloria B. Arkaifie, Ethel J. S. Blessie, Mohammed-Sherrif Fuseini, Isaac Carilo, Rebecca Yeboah, Leonard Asare, Brian D. Robertson

**Affiliations:** 1West African Center for Cell Biology of Infectious Pathogens, Department of Biochemistry, Cell and Molecular Biology, University of Ghana, P. O. Box LG 54, Accra, Ghana; vincentamarh02@gmail.com (V.A.); samalin01@gmail.com (P.C.); gloria.arkaifie@gmail.com (G.B.A.); jeblessie@gmail.com (E.J.S.B.); msherrif04@gmail.com (M.-S.F.); isaaccarilo@gmail.com (I.C.); rebecca.yeboah23@gmail.com (R.Y.); asaleonne@gmail.com (L.A.); 2Centre for Molecular Microbiology and Infection, Imperial College London, London SW7 2AZ, UK; b.robertson@imperial.ac.uk

**Keywords:** antibiotics, antimicrobial resistance, tuberculosis, mycobacteria

## Abstract

*Mycobacterium tuberculosis* is a pathogen of global public health concern. This threat is exacerbated by the emergence of multidrug-resistant and extremely-drug-resistant strains of the pathogen. We have obtained two distinct clones of multidrug-resistant *Mycobacterium smegmatis* after gradual exposure of *Mycobacterium smegmatis* mc^2^ 155 to increasing concentrations of erythromycin. The resulting resistant strains of *Mycobacterium smegmatis* exhibited robust viability in the presence of high concentrations of erythromycin and were found to be resistant to a wide range of other antimicrobials. They also displayed a unique growth phenotype in comparison to the parental drug-susceptible *Mycobacterium smegmatis* mc^2^ 155, and a distinct colony morphology in the presence of cholesterol. We propose that these two multidrug-resistant clones of *Mycobacterium smegmatis* could be used as model organisms at the inceptive phase of routine in vitro screening of novel antimicrobial agents targeted against multidrug-resistant *Mycobacterial tuberculosis*.

## 1. Introduction

Tuberculosis is one of the most successful human infections with an extensive global coverage [1]. It is caused by members of the *Mycobacterium tuberculosis* complex, following inhalation of aerosols containing the tubercle bacilli [2]. By estimation, one-third of the world’s population has latent tuberculosis infection and this represents a considerable reservoir from which future tuberculosis cases can emanate [3]. Currently, tuberculosis is a major cause of morbidity and death in low- and middle-income countries.

In the clinical setting, drug-resistant *Mycobacterium tuberculosis* is a serious and persistent threat to global health [4]. Drug resistance in *Mycobacterium tuberculosis* is largely associated with genetic mutations [5,6]. Resistance to at least rifampicin and isoniazid, also termed multidrug-resistant (MDR) tuberculosis, is on the rise. Over 490,000 MDR tuberculosis cases are recorded yearly, resulting in over 130,000 deaths [7]. Though host genetic factors may play a role, insufficient or incomplete treatment is the most important determinant of the development of MDR tuberculosis [8]. In the global setting, MDR tuberculosis accounts for 3.8% of new cases of the disease and presents in 20% of patients with a history of tuberculosis treatment [9].

Though a number of therapeutic strategies have been employed to treat and eradicate the disease, the emergence of drug-resistant strains has attracted worldwide concern. These strategies include the development of new drugs that target new bacterial proteins, cellular structures, and processes. Routine in vitro screening for novel antimycobacterial compounds would be facilitated by the availability of avirulent mycobacterial strains that are easy to grow and handle. Even though H37Ra is an avirulent strain of *Mycobacterium tuberculosis*, derived from the virulent H37 parent strain, its slow growth rate hinders the rapid initial screening of antimicrobials to identify candidates that exhibit antimycobacterial activity [10]. An alternative avirulent mycobacterial strain which could be used as a surrogate for *Mycobacterium tuberculosis* is the *Mycobacterium smegmatis* mc^2^ 155 strain; *Mycobacterium smegmatis* is suitable for rapid bioassays, which can be performed in less sophisticated biosafety cabinets. Notably, the diarylquinoline TMC207, which is an anti-tubercular drug, was identified using *Mycobacterium smegmatis* as a model organism [11]. This study set out to develop extensively-drug-resistant *Mycobacterium smegmatis* strains to provide a model which could be useful at the initial phase of screening for new antibiotics to combat drug-resistant infections.

Continuous exposure of microbial populations to antibiotics, as employed in this study, is considered the most relevant factor that influences the evolution of drug-resistant strains [12]. Continuous exposure to antibiotics promotes gradual acquisition of resistance by subpopulations of the mycobacterium, leaving their antibiotic susceptible counterparts extinct [13,14]. Interactions among other antibiotics also influence the extent of sensitivity of a microbe to a particular antibiotic. Antibiotic resistance has been shown to impose metabolic and fitness costs that reduces the growth of resistant strains [15,16].

Resistance to an antibiotic has been found to either promote cross-resistance or sensitivity to other antibiotics [17,18]. In the present study, drug-susceptible populations of *Mycobacterium smegmatis* mc^2^ 155 were driven, under in vitro conditions, to evolve into two distinct erythromycin-resistant strains. The study demonstrates the unique phenotypes exhibited by the two erythromycin-resistant strains, in comparison to the original mycobacterium strain from which they were derived. Collectively, we provide additional data which corroborate our current understanding of the emergence and characteristic phenotypes of drug-resistant strains of mycobacterium following continuous exposure to a drug. The two strains will also serve as valuable drug screening tools for the development of the next generation of anti-mycobacterial agents.

## 2. Materials and Method

### 2.1. Bacterial Strains, Media Preparation, and Antibiotic Treatments

*Mycobacterium smegmatis* mc^2^ 155 was used as a drug-susceptible strain for all experiments. The MDR *Mycobacterium smegmatis* strains (referred to as erythromycin-resistant *Mycobacterium smegmatis* A and B) were generated in this study, under in vitro conditions, and were used as drug-resistant strains. Middlebrook 7H10 agar base (Sigma Aldrich) and Middlebrook 7H9 broth base (Sigma Aldrich) were used for preparing agar plates and liquid broth, respectively, according to the manufacturer’s instructions. The M7H10 agar base was supplemented with 0.085% NaCl and 0.5% dextrose and the M7H9 broth base was supplemented with 0.085% NaCl, 0.44% glycerol, and 0.25% Tween 80. All commercially available antibiotics that were used for this study are listed in Appendix A. The amount or concentration of each antibiotic used for specific assays are indicated in the text of the results.

### 2.2. Drug Susceptibility Assay

Prior to a drug susceptibility assay, mycobacterial cells were streaked onto fresh M7H10 agar plates and incubated at 30 °C [19] for 48 h. The cells that grew on the agar plates were confirmed to be mycobacteria, via acid-fast staining, and were used for the inoculation of 50 mL of M7H9 broth base. The inoculated broths were incubated at 30 °C for 24 h with shaking (160 rpm). The bacterial cultures that were obtained were diluted to an optical density at 600 nm (OD_600_) of 0.7 in 50 mL of freshly prepared M7H9 broth base and incubated at 30 °C for 24 h with shaking. 

The 24 h, day two cultures were diluted to OD_600_ of 0.7 and were used for the uniform inoculation (spreading) of M7H10 agar plates. Paper discs containing the indicated amount of antibiotic were placed on the inoculated plates and the plates were incubated at 30 °C for 48 h. The zone of inhibition (in mm) around each antibiotic disc was measured and used to ascertain the susceptibility profile of the organism to the panel of antibiotics. 

### 2.3. Testing Cell Viability via Spot Test Assay

The 24-h, day two cultures were diluted to OD_600_ of 0.7 in 1 mL of M7H9 broth base containing the indicated concentration of erythromycin. The control for the diluted samples, indicated as untreated, did not contain erythromycin. The diluted mycobacterial cultures (± erythromycin) were incubated at 30 °C for 0 h, 3 h, and 6 h, and an aliquot of 5 μL was spotted on M7H10 agar plates after the indicated duration of incubation. The M7H10 agar plates containing the spots of the bacterial culture were incubated at 30 °C for 48 h. 

### 2.4. Growth Profile of Mycobacterial Cells

The 24-h, day two cultures were diluted to OD_600_ of 0.01 in 50 mL of M7H9 broth base. The diluted cultures were incubated at 30 °C, with shaking, and the OD_600_ were measured periodically at the indicated time intervals. 

### 2.5. Colony Morphology Assay

The 24-h, day two cultures were diluted to OD_600_ of 0.7 and aliquots of 5 μL were spotted on M7H10 agar plates containing 100 mΜ of cholesterol. The corresponding control M710 agar plates did not contain cholesterol. For each M7H10 agar plate, 5 μL of the diluted bacterial cultures were spotted at four positions, which were separated approximately 20 mm from each other. The agar plates containing the spots of the bacterial cultures were incubated at 30 °C for 48 h. Colonies that grew on the agar plates were confirmed to be mycobacterial cells by acid-fast staining. 

### 2.6. Staining of Mycobacterial Cells

A colony of cells that had grown on M7H10 agar plates were spread uniformly on microscope slides. Gram staining and acid-fast staining were performed to confirm that the cells were mycobacteria. Following the Gram staining and acid-fast staining, the cells were visualized using a Bresser Science MPO 401 light microscope equipped with a 100× objective lens. Images were acquired using a camera that was in-built within the microscope. 

## 3. Results

### 3.1. Exposure of Mycobacteria to Increasing Antibiotic Concentrations Drives Evolution of Drug Resistance

Starting with a single colony of *Mycobacterium smegmatis* mc^2^ 155, drug-resistant populations of the bacteria were allowed to evolve by systematically exposing the cells to increasing concentrations of either erythromycin, streptomycin, or tetracycline (Figure 1). Prior to each repetition (R), the minimal inhibitory concentrations (MICs) of erythromycin, streptomycin, and tetracycline were determined for the distinct colonies that had grown either within or around the zones of inhibition of these antibiotics (Figure 1). The MIC determination was used to ascertain the extent of the antibiotic resistance acquired by these distinct colonies of *Mycobacterium smegmatis*. The cells gradually adapted to the systematic exposure to increasing concentrations of each antibiotic, revealing gradual acquisition of resistance (Table 1). Measurement of OD_600_ at the end of the 2nd, 4th, and 5th steps also confirmed this observation (Appendix A). At the end of the 5th step, colonies of *Mycobacterium smegmatis* were not detected within the zone of inhibition on the M7H10 agar plates containing either streptomycin or tetracycline discs (Table 1). However, distinct colonies of *Mycobacterium smegmatis* were detected within the zones of inhibition of erythromycin discs on the M7H10 agar plates. These mycobacterial cells were characterized as erythromycin-resistant *Mycobacterium smegmatis*. Two colonies, isolated by spreading a 10^−6^ dilution of the erythromycin-resistant *Mycobacterium smegmatis* culture, showed different susceptibilities to a panel of thirteen commercial antibiotics (Figure 2a,b). These two distinct clones were further characterized as erythromycin-resistant *Mycobacterium smegmatis* A and B and were continually maintained on M7H10 agar plates (without antibiotics) for the next 6 months, prior to the subsequent assays. 

The *Mycobacterium smegmatis* mc^2^ 155 was susceptible to twelve out of the thirteen panel of commercial antibiotics, while the two clones of erythromycin-resistant *Mycobacterium smegmatis* (A and B) were both susceptible to only rifampicin, moxifloxacin, and streptomycin (Figure 2b). Notably, clone A, but not clone B, of the erythromycin-resistant *Mycobacterium smegmatis* strain was also susceptible to tetracycline and chloramphenicol. Hence, the distinct antibiotic susceptibility pattern for each erythromycin-resistant strain was the basis for the characterization of erythromycin-resistant *Mycobacterium smegmatis* as either clone A or clone B. These observations agree with a previous report that indicated that a single antibiotic treatment may select for resistance against other multiple drugs via a set of mechanisms (collateral resistance) or a single mechanism (cross-resistance) [20]. What is more striking about this data is that the acquisition of broad-spectrum resistance cuts across all classes of antibiotics. This observation is the first to be reported for mycobacteria where exposure to one antibiotic has led to the development of extreme drug resistance (MDR). Thus, these two strains constitute a valuable tool in the continued search for new antibiotic candidates.

### 3.2. MDR Mycobacterium Smegmatis is Viable in the Presence of Higher Concentrations of Erythromycin

A spot test assay was used to validate the extent of resistance of the two clones of erythromycin-resistant *Mycobacterium smegmatis* (clones A and B) relative to the *Mycobacterium smegmatis* mc^2^ 155 strain. These three mycobacterial strains were exposed to increasing concentrations of erythromycin for 0 h, 3 h, and 6 h, prior to spotting the cells on M7H10 agar plates. The erythromycin-resistant *Mycobacterium smegmatis* strains were viable at all of the three time points (0 h, 3 h, and 6 h of exposure to erythromycin), and the viability was robust against the low and high concentrations of erythromycin used for the assay (Figure 3). On the contrary, the *Mycobacterium smegmatis* mc^2^ 155 strain was viable at the low concentrations of erythromycin, whereas growth was undetected at the high concentrations of the antibiotic (Figure 3a,b). Notably, growth was undetected for the *Mycobacterium smegmatis* mc^2^ 155 strain following 6 h of exposure to the low concentrations of erythromycin (Figure 3c). These observations demonstrate that the two erythromycin-resistant strains exhibit a distinct viability phenotype in the presence of erythromycin, in comparison to the *Mycobacterium smegmatis* mc^2^ 155 parent strain.

### 3.3. MICs of Selected Antibiotics against MDR Mycobacterium Smegmatis 

The MICs of thirteen commercially-available antibiotics were determined for the *Mycobacterium smegmatis* mc^2^ 155 strain and the two MDR strains. For each organism, the MIC of an antibiotic of interest was determined by placing four discs, containing decreasing concentrations of the antibiotic, at positions distant from each other on the agar plate. All of the three mycobacterial strains were resistant to the highest concentration of Pyrazinamide (160 µg) and were susceptible to the lowest concentration of moxifloxacin (0.25 µg; Table 2). The two MDR strains were resistant to the highest concentration of isoniazid, ethambutol, linezolid, and erythromycin, while the *Mycobacterium smegmatis* mc^2^ 155 strain was susceptible to the lowest concentration of these antibiotics. The MIC of vancomycin and streptomycin was 80 µg for the MDR strains; the *Mycobacterium smegmatis* mc^2^ 155 strain was susceptible to the lowest concentrations of these antibiotics used for the assay. Interestingly, the two MDR strains were susceptible to 20 µg of ampicillin and amoxicillin, even though our earlier observations indicated that these MDR strains were resistant to 40 µg of either of these antibiotics (Figure 2b). Moreover, clone B of the MDR strain was susceptible to 15 µg and 30 µg of tetracycline and chloramphenicol, respectively; an observation that was inconsistent with the data shown in Figure 2b. We proposed that the placement of different classes of antibiotic discs on an agar plate during the test for the drug susceptibility pattern (Figure 2B), as compared to the placement of discs of the same antibiotic for the MIC determination (Table 2), was the basis for the above-mentioned deviations. The antibiotic assay format is the standard widely used for the determination of the antibiotic susceptibility pattern of bacterial isolates. Thus cross-interaction between different classes of antibiotics might be influencing the resistance profiles of the MDR *M. smegmatis* strains to a sub-collection of the selected commercial antibiotics. 

### 3.4. Effect of Neighboring Antibiotic Discs on the Antimycobacterial Activity of Selected Antibiotics 

Using the disc diffusion assay, a number of antibiotics are usually placed at unique positions on an agar plate that has been inoculated with the strain of interest during the test for drug susceptibility patterns (Figure 4). We investigated the effect of neighboring antibiotic discs on the antimycobacterial activity of ampicillin, amoxicillin, linezolid, tetracycline, chloramphenicol, erythromycin, and streptomycin. Ultimately, this experimental approach was anticipated to provide information on specific neighboring antibiotic discs that either enhanced or reduced the antimycobacterial activity of these selected antibiotic discs. 

The erythromycin-resistant *M. smegmatis* A strain was resistant to ampicillin, amoxicillin, and chloramphenicol when each of these selected antibiotics were surrounded by the neighboring antibiotic discs indicated in Table 3. Interestingly, the same concentration of these selected antibiotics (ampicillin, amoxicillin, and chloramphenicol) exhibited antimycobacterial activity against the same resistant strain in the absence of the neighboring antibiotic discs. The opposite effect was detected for streptomycin against the resistant strain when discs of rifampicin, isoniazid, and ampicillin were used as neighbors. These observations were not unique to erythromycin-resistant *M. smegmatis* A strain; the *M. smegmatis* mc^2^ 155 strain was highly susceptible to linezolid in the presence of neighboring antibiotic discs, but was resistant to the same concentration of linezolid in the absence of neighboring antibiotic discs. Collectively, these observations demonstrate that the resistance profiles of *M. smegmatis* mc^2^ 155 and the corresponding MDR strains are influenced by cross-interaction of the different classes of antibiotics. This same dataset demonstrates a subtle phenotypic difference between the two MDR strains.

### 3.5. Evolution of the MDR Phenotype Affects the Growth Profile of Mycobacterium Smegmatis

The growth profiles of the two clones of erythromycin-resistant *Mycobacterium smegmatis* and the *Mycobacterium smegmatis* mc^2^ 155 strain were investigated in order to ascertain the effect of the MDR phenotype on growth rate. Single colonies from these three mycobacterial strains were initially grown in M7H9 broth for 24 h at 30 °C with shaking. These mycobacterial cultures were diluted to an OD_600_ of 0.01 and the growth rate was monitored for 60 h (Figure 5a). The two clones of erythromycin-resistant *Mycobacterium smegmatis* showed a relatively rapid growth rate during the initial 12 h compared to the *Mycobacterium smegmatis* mc^2^ 155 strain. Interestingly, the latter strain recuperated from the initial lag phase of growth after 12 h and showed a rapid growth rate, attaining an OD_600_ of 3.2 by 48 h (Figure 5a). On the contrary, the erythromycin-resistant strains showed a relatively slower growth rate after the initial 12 h. This pattern of growth was confirmed following measurement of OD_600_ at 12-h intervals for 96 h (Figure 5b), illustrating that the acquisition of MDR phenotype causes a concomitant alteration in the growth profile of *Mycobacterium smegmatis*. 

### 3.6. Cholesterol Enhances the Growth of Colonies of Mycobacterium Smegmatis

It has been reported that *Mycobacterium tuberculosis* utilizes the cholesterol of its host organism, for virulence and invasion of macrophages, in a mechanism dependent on the modulation of lysosomal calcium levels [21,22]. Cholesterol metabolism in *M. tuberculosis* is facilitated by the presence of an operon, which consist of genes encoding acrylamine N-acetyltransferase and other associated proteins [23]. Proteins associated with cholesterol metabolism are usually conserved in *M. tuberculosis* and *M. smegmatis* and represent a possible target for novel anti-tubercular drugs [24,25]. Even though *M. smegmatis* is avirulent and does not persist in macrophages, the presence of these conserved proteins makes *M. smegmatis* a suitable laboratory model for studying the phenotypic effect of cholesterol on mycobacteria. Hence, we investigated the effect of cholesterol on the morphology of *Mycobacterium smegmatis* mc^2^ 155, as well as the erythromycin-resistant strains. 

Colonies of the *Mycobacterium smegmatis* mc^2^ 155 strain were detected as broad, fluffy-like structures on the cholesterol-containing M7H10 agar plates, whereas the control (no cholesterol) agar plates showed the expected colony phenotype (Figure 6a). The colonies of the two clones of erythromycin-resistant *Mycobacterium smegmatis* were modestly enlarged on the cholesterol-containing agar plate compared to the colonies on the control agar plates (Figure 6a). Visualization of individual cells within these colonies revealed that all of the three mycobacterial strains tested positive for acid-fast staining (Appendix A). Moreover, individual cells from colonies of the *Mycobacterium smegmatis* mc^2^ 155 strain had a slender and rod-like morphology (Figure 6b). Even though the erythromycin-resistant strains were also rod-like in shape, they were relatively smaller than the *Mycobacterium smegmatis* mc^2^ 155 strain. Importantly, the presence of cholesterol had no detectable effect on the morphology of the individual cells (Figure 6b). Hence, cholesterol enhances the spreading of the colonies of *Mycobacterium smegmatis* across the agar surface, of which we postulate might be an essential feature for tissue invasion by *Mycobacterial tuberculosis* if the phenotype is conserved across *Mycobacterium* sp.

## 4. Discussion

*Mycobacterium smegmatis* has been reported as a suitable surrogate *for Mycobacterium tuberculosis* during screening for novel anti-tuberculosis (anti-TB) drugs [26]. However, the emergence of MDR-tuberculosis (MDR-TB) and extensively-drug-resistant tuberculosis (XDR-TB) restrict the suitability of the drug-susceptible *Mycobacterium smegmatis* for the development of novel drugs against resistant TB strains. In the clinical settings, drug-resistant strains of TB emerge following the exposure of the drug-susceptible strain to mono-drug treatment conditions [27]. This strategy was explored in the present study for isolating two unique MDR strains of *Mycobacterium smegmatis*, under in vitro conditions. We have demonstrated that the drug-susceptible *Mycobacterium smegmatis* mc^2^ 155 strain is capable of evolving into distinct clones of MDR strains, following gradual exposure to increasing concentrations of erythromycin. The MDR *Mycobacterium smegmatis* (referred to as erythromycin-resistant *Mycobacterium smegmatis* A and B) were robust and viable in the presence of high concentrations of erythromycin. Interestingly, the MDR phenotype affected the growth profile of *Mycobacterium smegmatis*. This observation is not entirely surprising, as the ability of a drug-susceptible mycobacterium to adapt to increasing concentrations of an antibiotic has been linked to compensatory adaptations, which impacts fitness and metabolic costs [28]. The strategy of switching between broth and plate cultures and the use of antibiotic discs was critical in the isolation of these unique strains.

Antibiotics such as ampicillin and amoxicillin, which did not exhibit activity against the MDR *Mycobacterium smegmatis* strain, gained their activity in the presence of neighboring antibiotics, and vice versa. These observations illustrate the effect of antibiotic cross-interaction on the dynamic profile of the MDR phenotype. Even though combination chemotherapy has been well established as a robust strategy for eradicating MDR strains [29], our data also suggest that it is capable of suppressing or eliminating the activity of a useful antibiotic. Collectively, these provide insights into the intricate dynamics of drug resistance profiles that are exhibited by pathogenic bacteria, in general. 

According to the WHO, a strain of *Mycobacterium tuberculosis* is classified as MDR if it is resistant to at least the two most potent anti-TB drugs (rifampicin and isoniazid). Extremely-drug-resistant TB strains exhibit additional resistance to isoniazid, a fluoroquinolone, and one of the second-line anti-TB drugs (kanamycin, amikacin, and capreomycin). The two clones of the erythromycin-resistant *Mycobacterium smegmatis* generated by this study were both susceptible to rifampicin and a fluoroquinolone (moxifloxacin), but were resistant to isoniazid. Thus, the drug resistance profile of the erythromycin-resistant *Mycobacterium smegmatis* was not entirely consistent with the WHO criteria for the classification of MDR phenotypes of mycobacterium. The susceptibilities to moxifloxacin and streptomycin were markedly reduced in these new MDR strains, which somewhat place them between the classical MDR and XDR mycobacterial strains. Nonetheless, the MDR phenotype exhibited by the two distinct clones of erythromycin-resistant *Mycobacterium smegmatis* makes the organism a more suitable model organism for the identification of novel bioactive compounds against MDR-TB and XDR-TB, as compared to the *Mycobacterium smegmatis* mc^2^ 155 strain. Due to the significant genetic disparity between *Mycobacterium tuberculosis* and *Mycobacterium smegmatis*, the MDR strains of the latter would be more useful as model organisms at the initial phase of routine chemical library screening procedures. Future studies will be dedicated to the genomic and proteomic characterization of these MDR strains to unravel the mechanisms of resistance. Novel compounds that are selected as potential anti-TB drugs during screening of the chemical library using these MDR *Mycobacterium smegmatis* strains must undergo extensive characterization to validate their efficacy.

## 5. Conclusions

Continuous exposure of *Mycobacterium smegmatis* mc^2^ 155 to erythromycin generated two clones that were extremely resistant to isoniazid, ethambutol and linezolid, while exhibiting reduced susceptibility to other notable anti-TB drugs. The fast growth phenotype of *M. smegmatis* was maintained in these MDR strains, thereby consolidating their suitability as model organisms for rapid and routine in vitro screening of novel drugs targeted against MDR *Mycobacterium tuberculosis*.

## Figures and Tables

**Figure 1 antibiotics-08-00004-f001:**
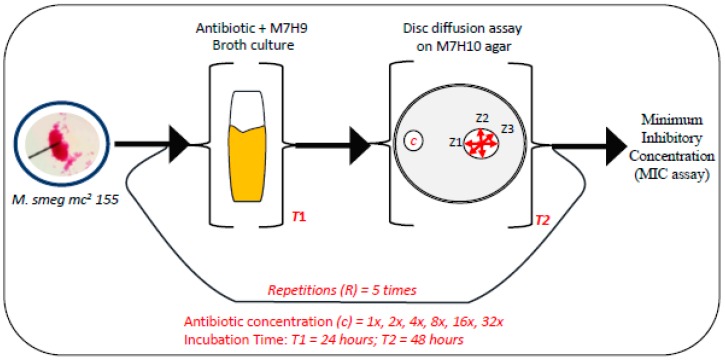
Generating multidrug-resistant (MDR) clones of *Mycobacterium smegmatis*, in vitro. Single colonies of the *Mycobacterium smegmatis* mc^2^ 155 strain were transferred into a M7H9 broth, containing a sublethal concentration of an antibiotic (streptomycin, erythromycin, or tetracycline), and incubated at 30 °C for 24 h with shaking. An aliquot of each broth culture was spread, uniformly, onto M7H10 agar plates and two paper discs containing the same amount of antibiotic (streptomycin, erythromycin, or tetracycline) were placed approximately 30 mm apart on the corresponding inoculated agar plate. All the agar plates were incubated at 30 °C for 48 h. Discrete colonies of *Mycobacterium smegmatis* that grew within the zones of inhibition were re-inoculated into fresh M7H9 broth for determination of the minimum inhibitory concentrations (MICs) of these colonies for each antibiotic. The discrete colonies were also inoculated into fresh M7H9 broth containing an increased concentration of antibiotic for repetition of the entire procedure (5 times). During the repetitions (R), the concentration of antibiotic that was added to M7H9 broth was increased systematically (2×, 4×, 8×, 16×, and 32×) prior to the disc diffusion assay. The initial concentrations of streptomycin, erythromycin, and tetracycline (1×), added to each fresh M7H9 broth, were 30 µg/µL, 60 µg/µL, and 30 µg/µL, respectively. Z1, Z2, and Z3 represent triplicate measurements of each zone of inhibition (in mm).

**Figure 2 antibiotics-08-00004-f002:**
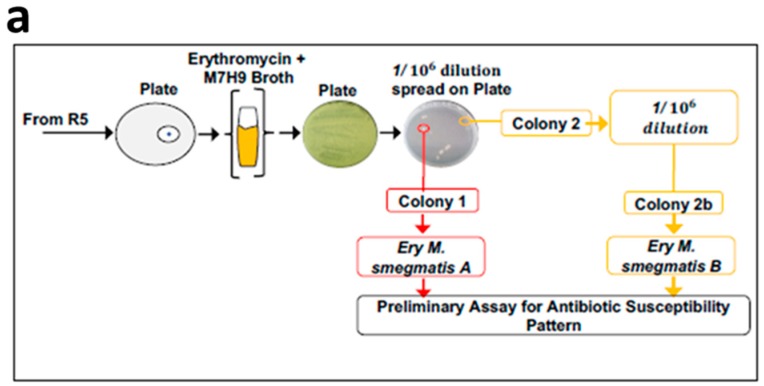
Isolation of two clones of MDR *Mycobacterium smegmatis*. (**a**) Schematic representation of the procedure used for isolating the two clones of erythromycin-resistant *Mycobacterium smegmatis*. Following the 5th repeat for generating resistant clones of *Mycobacterium smegmatis*, in vitro, (R5), a 10-fold serial dilution of the bacterial culture was performed. The 10^−6^ culture dilution was spread on a M7H10 agar plate, from which two distinct colonies of acid-fast bacilli were isolated and identified as erythromycin-resistant *Mycobacterium smegmatis* A and B. (**b**) Drug susceptibility pattern of the two clones of the erythromycin-resistant *Mycobacterium smegmatis*. The definitions of the abbreviated commercial antibiotics and their corresponding amounts in the discs are indicated in Appendix A.

**Figure 3 antibiotics-08-00004-f003:**
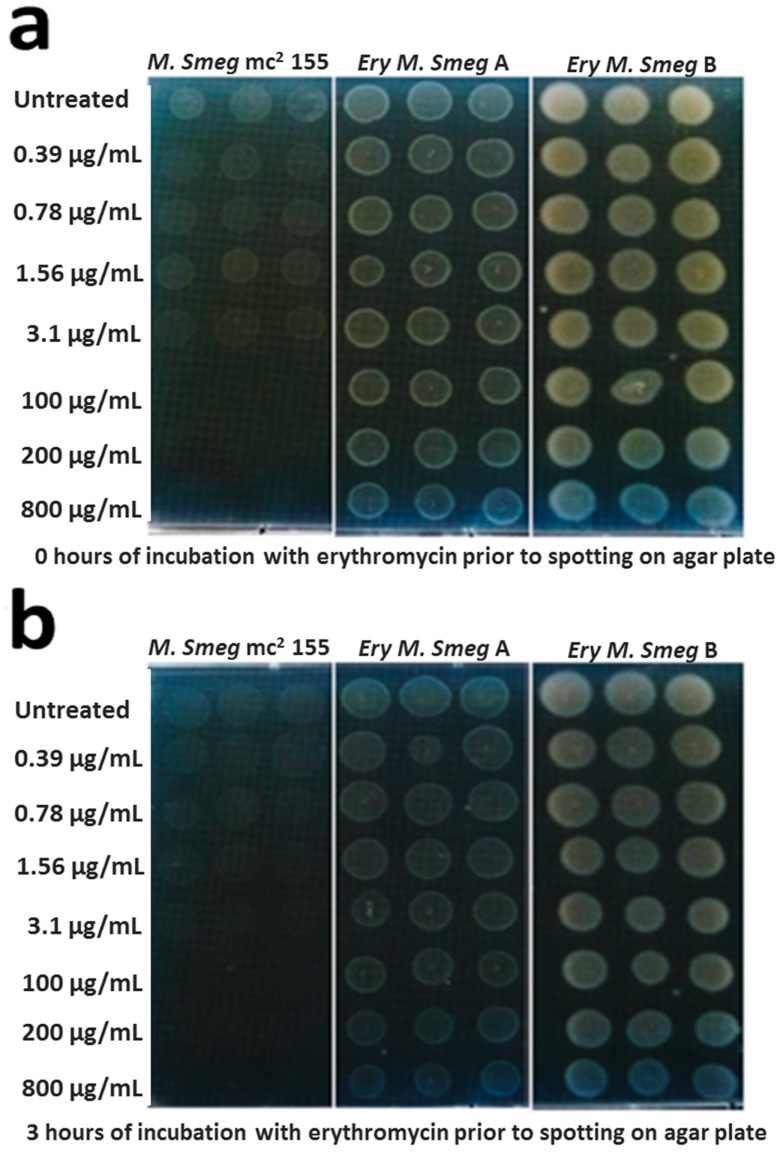
Viability assay of the drug-susceptible and MDR *Mycobacterium smegmatis* strains following exposure to low and high concentrations of erythromycin. Overnight cultures of these strains were either untreated or exposed to low (0.39 μg/mL; 0.78 μg/mL; 1.56 μg/mL; 3.1 μg/mL) or high (100 μg/mL; 200 μg/mL; 800 μg/mL) concentrations of erythromycin for either 0 h (**a**), 3 h (**b**), or 6 h (**c**). An aliquot of 5 μL of the untreated and treated cultures were spotted onto M7H10 agar plates and incubated at 30 °C for 48 h. For each treatment condition, the three spots on the M7H10 agar plate represent biological replicates. *Ery M. smeg A* and *Ery M. smeg B* represent the two erythromycin-resistant clones derived from the *Mycobacterium smegmatis* mc^2^ 155 strain.

**Figure 4 antibiotics-08-00004-f004:**
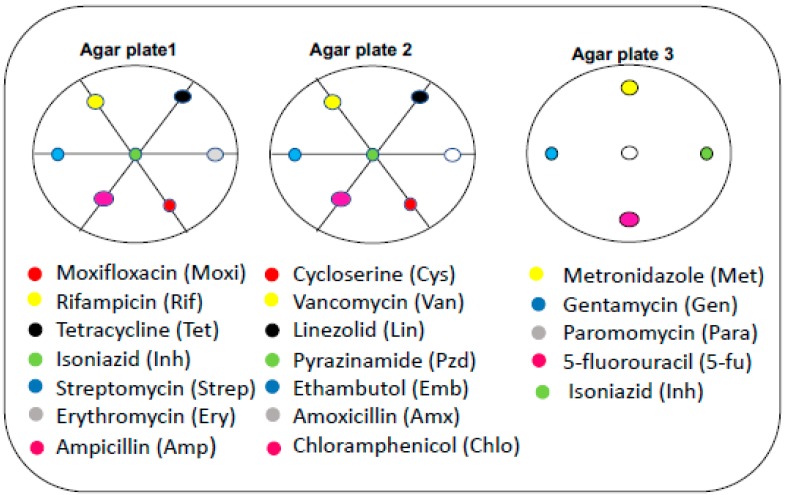
Schematic representation of the arrangement of antibiotic discs on agar plates. A total of nineteen different commercial antibiotics were placed at unique positions on three M7H10 agar plates, which were inoculated with cultures of either *Mycobacterium smegmatis* mc^2^ 155 or erythromycin-resistant *Mycobacterium smegmatis* A. The amount (in μg) of each selected antibiotic-containing disc is indicated in Appendix A and the amount of each antibiotic (in μg) used as a neighbor for the selected antibiotic is indicated in Appendix A.

**Figure 5 antibiotics-08-00004-f005:**
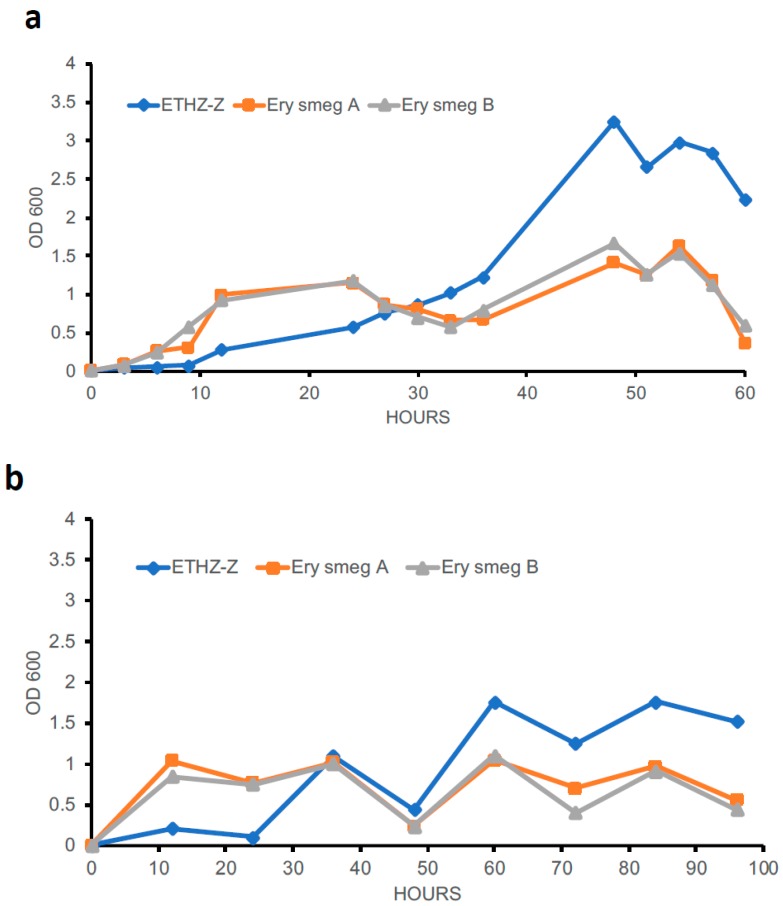
Growth profiles of the drug-susceptible and MDR *Mycobacterium smegmatis* strains. Overnight cultures of the three strains were diluted to an optical density at 600 nm (OD_600_) of 0.01 and the growth rate was monitored for 60 h (**a**) or 96 h (**b**). For (**a**), the OD_600_ was measured every 3 h for 12 h, followed by an intermittent 12-h measurement. For (**b**), the OD_600_ was measured every 12 h for 96 h. *Ery M. smeg* A and *Ery M. smeg* B represent the two erythromycin-resistant clones derived from the *Mycobacterium smegmatis* mc^2^ 155 strain.

**Figure 6 antibiotics-08-00004-f006:**
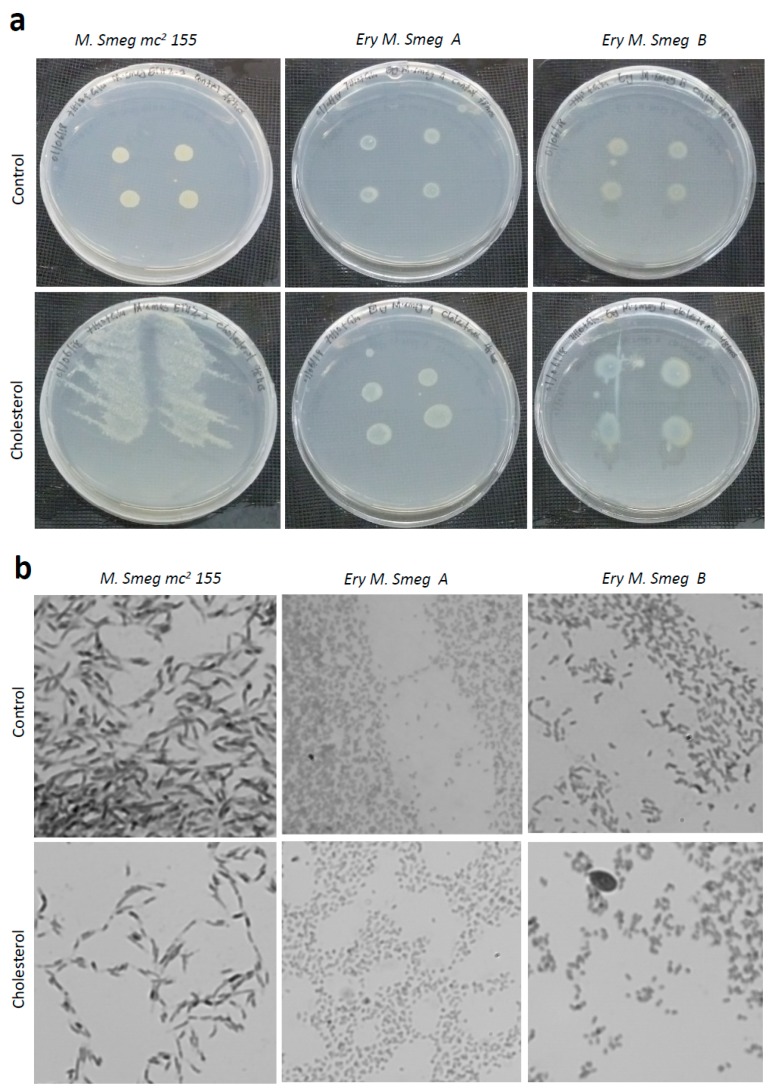
Effect of cholesterol on the morphology of colonies and individual cells of the drug-susceptible and MDR *Mycobacterium smegmatis* strains. (**a**) Colony morphology of the three *Mycobacterium smegmatis* strains in the absence and presence of cholesterol. Overnight cultures of the three strains were diluted to an OD_600_ of 0.7, and aliquots of 5 μL were spotted at four positions on M7H10 agar plates containing 100 mM cholesterol, as well as M7H10 agar plates that did not contain cholesterol (control). The plates were incubated for 48 h at 30 °C. (**b**) Visualization of individual cells from the mycobacterial colonies on the M7H10 agar plates, shown in (**a**). *Ery M. smeg A* and *Ery M. smeg B* represent the two erythromycin-resistant clones derived from the *Mycobacterium smegmatis* mc^2^ 155 strain.

**Table 1 antibiotics-08-00004-t001:** Zone of inhibition.

Zones of Inhibition (mm)															
Repetitions (R)	Streptomycin (µg/µL)					Erythromycin (µg/µL)					Tetracycline (µg/µL)				
	0.1	1	6	30	150	0.1	1	6	30	150	0.1	1	6	30	150
-	0	6	12	15	22	0	0	0	11.5	19.5	8	9.5	18	22	27.5
1	0	0	10	12	17.5	0	0	0	6	13.5	0	8	11	15.5	19
2	0	0	9	8	13	0	0	0	6	10	0	0	8.5	11	14.5
3	0	0	0	0	18	0	0	0	16.5	24.5	10	6	10	13	15
4	0	0	0	0	17	0	0	0	10	17.5	N/A	N/A	N/A	N/A	N/A
5	N/A	N/A	N/A	N/A	N/A	0	0	0	6	13	N/A	N/A	N/A	N/A	N/A

N/A denotes not available.

**Table 2 antibiotics-08-00004-t002:** Zones of inhibition.

Antibiotic (×)	Antibiotic Concentration	Zones of Inhibition (mm)		
		*M. smeg* mc^2^ 155	*Ery M. smeg* A	*Ery M. smeg* B
Amp (40 µg)	0.5×	0	12	10
	1×	6	11	14
	2×	9	17	14
	4×	18	21	20
Amx (40 µg)	0.5×	6	12	11
	1×	9	18	17
	2×	13	18	19
	4×	10	18	20
Van (40 µg)	0.5×	13	0	0
	1×	20	0	0
	2×	20	7	9
	4×	21	9	10
Inh (10 µg)	0.5×	8	0	0
	1×	29	0	0
	2×	37	0	0
	4×	40	0	0
Emb (10 µg)	0.5×	15	0	0
	1×	36	0	0
	2×	46	0	0
	4×	51	0	0
Pzd (40 µg)	0.5×	0	0	0
	1×	0	0	0
	2×	0	0	0
	4×	0	0	0
Moxi (0.5 µg)	0.5×	30	10	10
	1×	36	18	12
	2×	40	22	20
	4×	47	29	24
Rif (10 µg)	0.5×	0	18	12
	1×	8	20	16
	2×	8	23	19
	4×	11	25	21
Lin (5 µg)	0.5×	0	0	0
	1×	7	0	0
	2×	8	0	0
	4×	10	0	0
Tet (30 µg)	0.5×	46	20	7
	1×	40	20	9
	2×	42	23	14
	4×	54	27	18
Chlo (30 µg)	0.5×	14	9	0
	1×	18	9	7
	2×	22	15	10
	4×	31	22	13
Ery (40 µg)	0.5×	8	0	0
	1×	10	0	0
	2×	10	0	0
	4×	15	0	0
Strep (20 µg)	0.5×	18	0	0
	1×	20	0	0
	2×	31	6	6
	4×	38	8	9

The full definitions of the antibiotics are as follows: Ampicillin (Amp), Amoxicillin (Amx), Vancomycin (Van), Isoniazid (Inh), Ethambutol (Emb), Pyrazinamide (Pzd), Moxifloxacin (Moxi), Rifampicin (Rif), Linezolid (Lin), Tetracycline (Tet), Chloramphenicol (Chlo), Erythromycin (Ery), Streptoimycin (Strep).

**Table 3 antibiotics-08-00004-t003:** Selected antibiotics.

Selected Antibiotics	Selected Antibiotics in the Presence of Neighbors		Selected Antibiotics alone		Neighbors
	*M. smeg* mc^2^ 155	*Ery M. smeg* A	*M. smeg* mc^2^ 155	*Ery M. smeg* A	
Amp	N/A	0	N/A	11	Strep, Inh, Moxi
Amx	N/A	0	N/A	18	Lin, Pzd, Cys
Lin	50	N/A	0	N/A	Van, Pzd, Amx
Tet	N/A	7	N/A	20	Rif, Inh, Ery
Chlo	N/A	0	N/A	9	Emb, Pzd, Cys
Ery	20	N/A	10	N/A	Tet, Inh, Moxi
Strep	48	14	20	0	Rif, Inh, Amp

Numbers represent zones of inhibition (mm) of the selected antibiotics. The amount of antibiotics (µg) used for the assay are indicated in Appendix A.

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
