# Peer review of "Characterization of Two New Multidrug-Resistant Strains of Mycobacterium smegmatis: Tools for Routine In Vitro Screening of Novel Anti-Mycobacterial Agents"

_antibiotics, 2019, doi:10.3390/antibiotics8010004_

Round 1
Reviewer 1 Report
The authors have used the organism M. smegmatis as a model for M. tuberculosis. They should comment on the use of this organism. It is useful definitely but they need to make a comment about using it.
General when describing mycobacteria in general neither italics nor upper case should be used.
the authors have refered to their drug resistant clones as A and B. In order to avoid any confusion they should use lower case a and b for panels in figures.
the authors describe two clones in detail which were generated in response to exposure to erythromycin. Did they develop any drug resistant clones with either of the other antibiotics which they used.
Fig. 5 has the same data in two panels although the results come from two separate growth experiments . These data are effectively repeats as far as this referee can ascertain. Why are they shown as two separate panels?
Fig. 6 - the effects described in the text in relation to the top panels are not clear in the image.
the effect of cholesterol as a fuel refers to growth of intracellular mycobacteria e.g. M. tuberculosis. In contrast M. smegmatis does not grow inside cells.
The authos should quote two additional papers in relation to cholesterol in mycobacteria inside macrophage
viz.
1)Ryan, et al British J Pharmacol. 2017 174:2209-2224. doi: 10.1111/bph.13810.
2)Fineran et al.,Wellcome Open Research 2016, 1:18 (doi:10.12688/wellcomeopenres.10036.1).
Author Response
Authors’ response to comments of reviewer #1
1. The authors have used the organism M. smegmatis as a model for M. tuberculosis. They should comment on the use of this organism. It is useful definitely but they need to make a comment about using it.
Response: We appreciate the fact that the reviewer agrees M. smegmatis is a useful model for M. tuberculosis. We have included additional texts at the third paragraph of the introduction to highlight/explain the use of M. smegmatis as a model for M. tuberculosis.
2. General when describing mycobacteria in general neither italics nor upper case should be used.
Response: We have removed the italics and upper case for instances where ‘mycobacteria’ was used (in general) throughout the text.
3. The authors have referred to their drug resistant clones as A and B. In order to avoid any confusion they should use lower case a and b for panels in figures.
Response: We have converted A and B for panels in the figures to a and b.
4. The authors describe two clones in detail which were generated in response to exposure to erythromycin. Did they develop any drug resistant clones with either of the other antibiotics which they used?
Response: Three antibiotics (streptomycin, erythromycin and tetracycline) were used at the initial phase of the study to drive evolution of resistant clones for these antibiotics (described at section 3.1 of the results and the legend for figure 1). However, the clones obtained from the usage of streptomycin and tetracycline had to be abandoned because they did not display consistent and robust resistance towards these two antibiotics (Table 1). We also decided to keep the preliminary results for streptomycin and tetracycline included in the data to provide our reviewers and readers the opportunity to appreciate the relatively strong performance of erythromycin in this context.
5. Fig. 5 has the same data in two panels although the results come from two separate growth experiments. These data are effectively repeats as far as this referee can ascertain. Why are they shown as two separate panels?
Response: The authors agree with the reviewer that Fig. 5a and 5b are repeats (mainly), however, there is a difference in the cultures presented. In the 5b, the cultures were not opened for the initial 12 hours to measure OD, whilst in the 5a, the culture bottles were opened every three hours to measure OD for the first 12 hours. It appears the initial and subsequent intermittent opening of the culture bottle every three hours boosted the overall growth profile of the WT strain.
6. Fig. 6 Â the effects described in the text in relation to the top panels are not clear in the image.
Response: For Fig. 6a, our data indicate that the presence of cholesterol on the M7H10 agar plate had an effect on the colony morphology formed by the three M. smegmatis strains. Specifically, the M. smegmatis mc2 155 strain exhibited enlarged fluffy-like morphology on the cholesterol-containing agar plate, but not on the control agar plate; the same volume of mycobacterial culture was spotted onto the cholesterol-containing agar plate and the control agar plate. Even though the Ery M. smegmatis A and B clones did not exhibit a fluffy-like morphology on the cholesterol-containing agar plate, we observed modest increase in the colony size in comparison to the colonies on the control agar plates. We believe this explanation describes our observation for Fig. 6a, as provided in the text of the 2nd paragraph of section 3.6 (results).
7. The effect of cholesterol as a fuel refers to growth of intracellular mycobacteria e.g. M. tuberculosis. In contrast M. smegmatis does not grow inside cells. The authors should quote two additional papers in relation to cholesterol in mycobacteria inside macrophage viz.
i) Ryan, et al British J Pharmacol. 2017 174:2209Â2224. doi: 10.1111/bph.13810.
ii) Fineran et al., Wellcome Open Research 2016, 1:18 (doi:10.12688/wellcomeopenres.10036.1).
Response: The authors have provided additional information at the first paragraph of the section 3.6 (results), which explains the rationale for using M. smegmatis to study the effect of cholesterol on mycobacteria. The two references proposed by the reviewer, together with other additional references, have been cited at the first paragraph of section 3.6 (results).
Reviewer 2 Report
General Comments:
Tuberculosis [TB] is a major global health concern. TB becomes even more complicated with the increasing rise of Multi drug resistant TB strains. The manuscript highlights the development of a novel tool to screen novel anti-mycobacterial agents by using the the two antibiotic resistant strain of M. smegmatis. The authors propose to use the mutant strain which shall mimic the virulent M. tuberculosis. Thereby, provide a safe and easy strategy to conduct drug screening against MDR-TB. The authors have a strong hypothesis of using the resistant M. smeg strain but data fail to convince their conclusions.
The manuscript scientific language and sections are well written but I have few specific questions as mention below.
Specific Comments:
Line 21-23: The authors fail to comprehend the use of M. smeg instead of Mtb. There was no introduction provided to the readers about the advantages of M. smeg.
Line 35-36: What is the genetic similarities between Mtb and M.smeg? Why did the authors choose H37Ra which is avirulent and closer to Mtb H37Rv?
Line 42-44: The authors started the introduction about Mtb and suddenly introduced M.smeg. It is a bit confusing for the reader. Can the author explain about the virulence properties of both the strain?
Line 58-60: What additional insights was provided?
Line 65: Why did the authors mentioned the two strain as MDR-M.smeg?
Line 76: In general, mycobacteria (especially Mtb) are grown at 37C which is also a human host body temperature. So, what was the reason for growing M. smeg at 30C?
Line 149: The data in Figure 3 does not look convincing at all. First, the authors ignore the observation found with the control untreated samples. It is clear that the WT M. smeg did not grow well in the untreated spots. What could be the possible reason? Second, there was a difference between the two resistant strains though there was no gradient reduction in growth. This show that the antibiotic was ineffective at a very high concentration.
Line 225: The virulent Mtb is known to utilize host cholesterol but since M. smeg is non-virulent
and does not survive in the host, so do the author think that this is significant to study? Also Figure 6, the control WT plate have a spread out growth instead of the spot? Or there is some problem with the picture?
Line 374: The authors name is in Upper case. Is it a typo ?
Author Response
Authors’ response to comments of reviewer #2
General Comments: Tuberculosis [TB] is a major global health concern. TB becomes even more complicated with the increasing rise of Multi drug resistant TB strains. The manuscript highlights the development of a novel tool to screen novel antimycobacterial agents by using the two antibiotic resistant strain of M. smegmatis. The authors propose to use the mutant strain which shall mimic the virulent M. tuberculosis. Thereby, provide a safe and easy strategy to conduct drug screening against MDRÂTB.
1. The authors have a strong hypothesis of using the resistant M. smeg strain but data fail to convince their conclusions.
Response: The hypothesis for this study was that the development of a MDR M. smegmatis strain would be a useful tool for routine in vitro screening of compounds to identify novel antimycobacterial compounds exhibiting activity against MDR TB strains. The aim of this study was therefore to develop these MDR M. smegmatis strains, which our data shows was successfully achieved.
The use of these MDR M. smegmatis strains to screen for novel antimycobacterial compounds is a study we are currently planning to initiate. It was not intended that such data would be provided in the current manuscript. This explanation is highlighted by the last sentence of the introduction. We also think that the use of a panel of antimycobacterial compounds demonstrates the critical difference in the drug susceptibility profile of the MDR M. smegmatis strains compared to the wild type.
The manuscript scientific language and sections are well written but I have few specific questions as mention below.
Specific Comments:
2. Line 21Â-23: The authors fail to comprehend the use of M. smeg instead of Mtb. There was no introduction provided to the readers about the advantages of M. smeg.
Response: The authors have added additional texts to the third paragraph of the introduction to highlight the advantages of using M. smegmatis as a model organism for Mtb.
3. Line 35Â-36: What is the genetic similarities between Mtb and M. smeg? Why did the authors choose H37Ra which is avirulent and closer to Mtb H37Rv?
Response: Even though H37Ra is an avirulent M. tuberculosis strain and closer to the virulent H37Rv strain, M. smegmatis was chosen over H37Ra because of the fast growth rate of M. smegmatis, which makes it suitable for performing rapid initial bioassays to identify potential novel anti-tubercular drugs. The authors are aware that Altaf et al., (2010) have published data indicating that only 50% of M. tuberculosis inhibitors are detected by M. smegmatis during chemical library screening. Moreover, the avirulence of M. smegmatis and its poor survival in macrophages indicate that M. smegmatis is quite genetically diverse from M. tuberculosis. However there exist key cellular processes that are conserved across these two mycobacterial strains, which could be exploited for development of novel chemotherapy. A typical example is the anti-tubercular drug, diarylquinoline TMC207, which targets the ATP synthase of M. tuberculosis; TMC207 was initially identified using M. smegmatis as a model organism for M. tuberculosis. Thus, the authors believe there is trade-off between using H37Ra (which is genetically very similar to M. tuberculosis, grows at a slower rate and thus does not favour rapid screening of chemical libraries) and M. smegmatis (which has a faster growth rate, favours rapid screening of chemical libraries and exhibits less sensitivity to detection of novel anti-tubercular drugs).
4. Line 42Â-44: The authors started the introduction about Mtb and suddenly introduced M. smeg. It is a bit confusing for the reader. Can the author explain about the virulence properties of both the strain?
Response: The authors have rearranged the text in the third paragraph of the introduction to facilitate a smooth transition from the usage of M. tuberculosis to M. smegmatis.
5. Line 58-Â60: What additional insights was provided?
Response: At the penultimate sentence of the introduction, the authors have replaced ‘…provide additional insights to the current understanding of the ….’ with ‘…provide additional data which corroborate our current understanding of the …..’
6. Line 65: Why did the authors mentioned the two strain as MDR ÂM.smeg?
Response: Amongst the panel of commercial antibiotics tested, the original M. smegmatis mc2 155 strain was resistant to only pyrazinamide while the two erythromycin-resistant M. smegmatis strains were resistant to isoniazid, pyrazinamide, ethambutol (which are first line anti-TB drugs) and linezolid (which is a third line anti-TB drug). These observations led us to refer to the two erythromycin-resistant strains as MDR M. smegmatis.
7. Line 76: In general, mycobacteria (especially Mtb) are grown at 37C which is also a human host body temperature. So, what was the reason for growing M. smeg at 30C?
Response: We chose 30oC for the reason that its closer to the growth conditions in the soil where it was isolated from. This is likely to make the cells more resilient to antibiotics. There are other studies that have also used 30oC (Noens et al., 2011). We plan to explore the impact of culturing of the cells at 30oC and 37oC on their antibiotic susceptibility.
8. Line 149: The data in Figure 3 does not look convincing at all. First, the authors ignore the observation found with the control untreated samples. It is clear that the WT M. smeg did not grow well in the untreated spots. What could be the possible reason? Second, there was a difference between the two resistant strains though there was no gradient reduction in growth. This show that the antibiotic was ineffective at a very high concentration.
Response: From our experience with working with these three M. smegmatis strains, the Ery M. smeg B strain shows a rapid and intense growth on the M7H10 agar plate, followed the Ery M. smeg A strain; the WT strain shows a relatively slow and faint growth on the agar plate. This observation prompted us to perform the growth curve shown in Figure 5. Currently, we do not have explanations for these differences in growth pattern on M7H10 agar plates. However, we are performing whole genome sequencing of these three strains to identify possible gene mutations in the two MDR strains which might explain their fast/intense growth on the agar plate.
An ineffective antibiotic at the higher concentrations would have led to growth of the WT strain at these high antibiotic concentrations. However, this was not observed on the agar plates shown in Figure 3.
9. Line 225: The virulent Mtb is known to utilize host cholesterol but since M. smeg is nonÂvirulent and does not survive in the host, so do the author think that this is significant to study? Also Figure 6, the control WT plate have a spread-out growth instead of the spot? Or there is some problem with the picture?
Response: Even though the avirulent M. smegmatis does not survive in the host’s macrophages, proteins that are utilized for cholesterol metabolism are conserved between M. tuberculosis and M. smegmatis. Moreover, these proteins have been suggested to represent potential targets for novel anti-tubercular drugs. Thus, the authors believe M. smegmatis can be used as a suitable laboratory model for studying phenotypic effects of cholesterol on mycobacteria.
These explanations have been incorporated into the first paragraph of section 3.6 (results) to clarify our use of M. smegmatis to study phenotypic effect of cholesterol on mycobacteria.
The authors were initially surprised by the unique morphology exhibited by the M. smegmatis mc2 155 strain on the cholesterol-containing agar plate. However, the same observation was made when the experiment was repeated. Acid-fast staining of cells from these spread-out colonies revealed that they were baccili-shaped and were positive for acid-fast staining. These observations indicate that the spreading out of the M. smegmatis mc2 155 strain on the cholesterol containing agar-plates was not an artefact.
10. Line 374: The authors name is in Upper case. Is it a typo?
Response: The authors name in upper case is a typo and we have corrected that.

Round 2
Reviewer 2 Report
In this revised submission, the authors made necessary corrections as per reviewer's comment. The authors also addressed most of the questions raised by the reviewer. But, it is unfortunate and still not convincing why the authors found a difference in growth pattern between the WT and mutant M. smeg. [Fig 3] This variation in physiology of the strain may or may not be linked to genetic variation as postulated by the authors. Since the authors based their model on 50% genetic similarity between the fast growing M. smeg and Mtb or H37Ra, it is important to have better insight on the efficacy of any drugs to be tested.
I believe that the authors will agree that 37C would be the best condition to mimic Mtb lifestyle and adaption in human infection. Don't you think there are important gene regulation differences at the two temperature mentioned by the authors?
Author Response
Response to 2nd round of comments from reviewer #2
1. In this revised submission, the authors made necessary corrections as per reviewer's comment. The authors also addressed most of the questions raised by the reviewer. But, it is unfortunate and still not convincing why the authors found a difference in growth pattern between the WT and mutant M. smeg. [Fig 3] This variation in physiology of the strain may or may not be linked to genetic variation as postulated by the authors. Since the authors based their model on 50% genetic similarity between the fast growing M. smeg and Mtb or H37Ra, it is important to have better insight on the efficacy of any drugs to be tested.
Response: The authors agree with the reviewer that the variation in growth phenotype between the WT and mutant M. smegmatis might not be linked to genetic variations. Our postulation is based on the outcomes of in vitro studies [1], in vivo studies [2] and clinical studies [3] where it had been reported that fitness cost arising from drug resistant mutations may be compensated by secondary mutations that enhances fitness without affecting the resistance phenotype [4]. We decided to report this observation (Figure 3) because it was consistently detected while conducting the study.
The authors agree with the reviewer that it is mandatory to obtain in-depth insight on the efficacy of novel drugs that are selected from screening of chemical library using M. smegmatis as a model organism. The authors acknowledge that the diarylquinoline class of drugs (identified from chemical screening using M. smegmatis) underwent extensive characterization to validate its efficacy as an anti-TB drug. Thus, we have added the following sentence (as a concluding remark) to the discussion section of the manuscript: ‘Novel compounds that are selected as potential anti-TB drugs during screening of chemical library using these MDR Mycobacterium smegmatis strains must undergo extensive characterization to validate their efficacy’.
2. I believe that the authors will agree that 37C would be the best condition to mimic Mtb lifestyle and adaption in human infection. Don't you think there are important gene regulation differences at the two temperature mentioned by the authors?
Response: The author’s completely agree with the reviewer that the two different temperatures (30°C compared to 37°C) can differentially affect regulation of gene expression. We also agree that 37°C is the optimal temperature which mimics Mtb lifestyle and adaptation during human infection. We are grateful for this comment and we shall take that into consideration in our current and future studies.
References:
1. Levin, B.R.; Perrot, V.; Walker, N. Compensatory mutations, antibiotic resistance and the population genetics of adaptive evolution in bacteria. Genetics 2000, 154, 985-997.
2. Bjorkman, J.; Nagaev, I.; Berg, O.G.; Hughes, D.; Andersson, D.I. Effects of environment on compensatory mutations to ameliorate costs of antibiotic resistance. Science 2000, 287, 1479-1482.
3. Gagneux, S.; Long, C.D.; Small, P.M.; Van, T.; Schoolnik, G.K.; Bohannan, B.J. The competitive cost of antibiotic resistance in Mycobacterium tuberculosis. Science 2006, 312, 1944-1946, doi:10.1126/science.1124410.
4. Melnyk, A.H.; Wong, A.; Kassen, R. The fitness costs of antibiotic resistance mutations. Evol Appl 2015, 8, 273-283, doi:10.1111/eva.12196.
